# A Robust Deep Learning Approach for Spatiotemporal Estimation of Satellite AOD and PM$_{2.5}$

**Lianfa Li** [1,2,3] 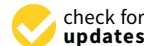

[1] State Key Laboratory of Resources and Environmental Information Systems, Institute of Geographic Sciences and Natural Resources Research, Chinese Academy of Sciences, Datun Road, Beijing 100101, China; lilf@lreis.ac.cn; Tel.: +86-10-648888362
[2] University of Chinese Academy of Sciences, Beijing 100049, China
[3] Spatial Data Intelligence Lab Ltd. Liability Co., Casper, WY 82609, USA

**Abstract:** Accurate estimation of fine particulate matter with diameter ≤2.5 μm (PM$_{2.5}$) at a high spatiotemporal resolution is crucial for the evaluation of its health effects. Previous studies face multiple challenges including limited ground measurements and availability of spatiotemporal covariates. Although the multiangle implementation of atmospheric correction (MAIAC) retrieves satellite aerosol optical depth (AOD) at a high spatiotemporal resolution, massive non-random missingness considerably limits its application in PM$_{2.5}$ estimation. Here, a deep learning approach, i.e., bootstrap aggregating (bagging) of autoencoder-based residual deep networks, was developed to make robust imputation of MAIAC AOD and further estimate PM$_{2.5}$ at a high spatial (1 km) and temporal (daily) resolution. The base model consisted of autoencoder-based residual networks where residual connections were introduced to improve learning performance. Bagging of residual networks was used to generate ensemble predictions for better accuracy and uncertainty estimates. As a case study, the proposed approach was applied to impute daily satellite AOD and subsequently estimate daily PM$_{2.5}$ in the Jing-Jin-Ji metropolitan region of China in 2015. The presented approach achieved competitive performance in AOD imputation (mean test R$^2$: 0.96; mean test RMSE: 0.06) and PM$_{2.5}$ estimation (test R$^2$: 0.90; test RMSE: 22.3 μg/m$^3$). In the additional independent tests using ground AERONET AOD and PM$_{2.5}$ measurements at the monitoring station of the U.S. Embassy in Beijing, this approach achieved high R$^2$ (0.82–0.97). Compared with the state-of-the-art machine learning method, XGBoost, the proposed approach generated more reasonable spatial variation for predicted PM$_{2.5}$ surfaces. Publically available covariates used included meteorology, MERRA2 PBLH and AOD, coordinates, and elevation. Other covariates such as cloud fractions or land-use were not used due to unavailability. The results of validation and independent testing demonstrate the usefulness of the proposed approach in exposure assessment of PM$_{2.5}$ using satellite AOD having massive missing values.

**Keywords:** PM$_{2.5}$; satellite AOD; deep learning; autoencoder; residual network; exposure estimation; high spatiotemporal resolution

## 1. Introduction

Fine particulate matter with a diameter < 2.5 μm (PM$_{2.5}$) has been associated with adverse health effects, ranging from acute, short-term to chronic health outcomes [1–3], including increased respiratory symptoms [4–6], worsened asthma [7,8], increased cardiovascular diseases [9,10], decreased lung function [11], and increased premature death from heart or lung diseases [12,13]. With rapidly increasing fleet of vehicles and more stringent emission regulations for industry than previously [14], concentrations of PM$_{2.5}$ have been increasingly more affected by local traffic emissions than coal or

industry emissions (e.g., an increase of 24% for vehicle emission but a decrease of 28.5% between 2014 and 2018 for industry and coal burning in Beijing of China [15,16], Figure S1). Thus, traffic emissions recently have contributed more to $PM_{2.5}$ concentrations, probably resulting in high exposure since many persons live near traffic routes [17–19]. Accurate estimation of the spatiotemporal distribution of $PM_{2.5}$ [20–22] at a high resolution can help improve health outcome studies of $PM_{2.5}$ [8,23], particularly at a local spatial scale. However, $PM_{2.5}$ ground monitoring stations are usually limitedly distributed worldwide, e.g., there were 102 $PM_{2.5}$ monitoring stations in 2015 for the large Jing-Jin-Ji metropolitan area of China (64,022 $km^2$). These limited measurements create a challenge in spatiotemporal $PM_{2.5}$ estimation at a high resolution using traditional modeling methods such as nearest neighbor [24,25], land-use regression [26,27], generalized additive model (GAM) and kriging [28].

Satellite-derived aerosol optical depth (AOD) has been used to predict $PM_{2.5}$ concentrations. AOD is a metric of the extinction of the solar beam by dust and haze and is widely used as a proxy for the number of particles within a vertical column and as a predictor of primary interest (assumed to account for the majority of the variance explained) for $PM_{2.5}$ [29]. Chemical transport models (CTMs), e.g., GEOS-Chem [30] and CMAQ [31], have also been used to simulate $PM_{2.5}$. The simulation of $PM_{2.5}$ is constrained by the incompleteness or unreliability of the input variables (e.g., emission inventory and meteorological parameters) and complex environmental processes. Consequently, CTMs may not predict well spatiotemporal $PM_{2.5}$ concentrations [32,33]. Since approximately the early 20th century, the moderate resolution imaging spectroradiometer (MODIS) onboard the polar orbiting Terra (launched in 1999) and Aqua (launched in 2002) satellites have been producing satellite AOD with global coverage. As two early widely-used mainstream methods to retrieve satellite AOD from MODIS, the dark target (DT) and deep blue (DB) algorithms [34,35] were developed by National Aeronautics and Space Administration (NASA). The DT algorithms have taken advantage of bright aerosols against a dark surface (hence named "dark target") to separate the aerosol and surface signals, and just worked over the dark vegetated targets over land and ocean, but not over bright land surface. Later, where the surface is bright and the DT doesn't work, the DB algorithm [36,37] has been developed to use the surface reflectance in the blue bands (hence named "deep blue") to capture the surface signals and spectral reflectance ratios over bright land surfaces. Until now, both complementary algorithms just have coarse spatial resolution (3–10 km) products.

Empirical statistical methods of correlation analysis [38–43] were initially used to model the AOD-$PM_{2.5}$ linear relationship with limited accuracy or resolution. Then, with satellite AOD as a predictor of primary interest, advanced approaches were applied to predict $PM_{2.5}$. For example, Sorek-Hamer et al., 2013 used non-parametric additive models to construct non-linear relationships between $PM_{2.5}$ and covariates [44], Lee et al., 2011 and Xie et al., 2015 introduced fixed and random effects in the mixed-effects models, allowing for daily variations of the AOD-$PM_{2.5}$ relationship [45,46], Li et al., 2017 used nonparametric generalized regression neural networks through normalized radial basis functions to predict $PM_{2.5}$ concentrations [47], Zhan et al., 2017 modeled spatial non-stationarity in the AOD-$PM_{2.5}$ association using geographically-weighted gradient boosting machines [48], and Guo et al., 2017 integrated spatial and temporal information in geographically and temporally weighted regression to capture $PM_{2.5}$ spatiotemporal variations in local effects [49]. However, moderate to coarse spatial resolution AOD products were unable to produce high spatial resolution $PM_{2.5}$ estimates.

Recently, the multiangle implementation of atmospheric correction (MAIAC) has been used as an advanced algorithm to retrieve AOD according to the time series of MODIS measurements at a high spatial (1 km) and temporal (daily) resolution, with better atmospheric correction over both dark and moderately bright surfaces, compared with that of the DT and DB algorithms [50–52]. Based on the MAIAC AOD, Xu et al. [53] developed two-stage models (mixed-effect model and geographically weighted regression (GWR)) to estimate $PM_{2.5}$ at a high resolution with cross-validation (CV) $R^2$ of 0.67. However, for either AOD retrieved by DT/DB or by MAIAC, a primary limitation is the massive non-random missingness caused by clouds, snow and conditions of high surface reflectance (e.g., an average missing percentage of > 60% in 2013-2014 in the Yangtze River Delta of China [54]).

Such missing data create a challenge for using AOD in $PM_{2.5}$ prediction since missing AOD may result in significant bias in the estimation of $PM_{2.5}$ [55]. Two studies used the AODs in neighboring cells [56] or CTM outputs [57] as alternatives to missing AOD values. These relatively simple methods may not perform well when a large amount (e.g., >50%) of the AOD data is missing. Other studies imputed missing AOD values using CTM output, meteorology, and elevation, etc., through different models including a two-step model (linear mixed model and kriging) [58], non-linear additive models [54] or neural networks [59]. However, these studies only had poor to moderate (mean $R^2$: 0.18–0.44) validation results when compared to AOD measurements from the aerosol robotic monitoring networks (AERONET).

The quality of covariates (e.g., completeness and measurement accuracy) will substantially influence model performance. In addition, missing covariates at certain locations or time points may make certain methods inapplicable. For example, cloud fraction data have been used by previous studies [60,61], but they were not available in earlier years, which might result in a substantial reduction in the performance in the GAM-based imputation method [54].

Machine learning methods, such as neural networks [59], random forest [57] and bootstrap aggregating (bagging) of mixed-effect models [62], are increasingly applied in air pollutant exposure assessment. However, a regular feed-forward neural network may have issues related to saturation and degradation of accuracy with increased hidden layers, as shown in many other applications [63,64]. Decision tree-based machine learning methods, such as random forest and XGBoost [65], are good in categorical variable classification but not ideal for continuous variables as they may produce abrupt changes in spatial variation in prediction due to the discretization of continuous covariates in these methods. Further, except for ensemble methods, such as random forest, most of the other methods are based on the single-model method where a single value is predicted by a trained model [62]. Sampling bias in training may result in unstable predictions with a large variance [66].

Although deep learning has achieved great successes in many domains [67], its applications involving the regression of a continuous variable, such as the imputation of missing MAIAC AOD and estimation of $PM_{2.5}$, are limited due to low efficiency in learning for neural networks. Many applications involving image classification and segmentation effectively use the convolutional neural network (CNN) [68]. However, CNN cannot work well with samples that contain massive non-random missing data of continuous variables [67] and involve complex environmental processes, and thus CNN was not considered in this paper.

This paper aims to present a novel deep learning approach for robust spatiotemporal imputation of MAIAC AOD and subsequent spatiotemporal modeling of $PM_{2.5}$ at a high (1-km) resolution. Here, an autoencoder-based residual deep network (also referred to as residual network throughout this paper) can be used to extract latent representation, implement residual connections, and share parameters in the multivariable output, which can considerably boost efficiency and accuracy of learning. Bagging of residual networks can further improve model performance by generating stable ensemble predictions with uncertainty estimates. Experiments are conducted to demonstrate the performance of the proposed approach, compared with other machine learning methods, and the implication for AOD imputation and $PM_{2.5}$ modeling is discussed.

## 2. Materials

### 2.1. Study Region

The study region (Figure S2) covers most of the Jing-Jin-Ji metropolitan region of China, located between the latitudes of 38°05′N and 41°04′N and the longitudes of 115°14′E and 118°54′E, with an area of 64,022 km$^2$ and a total population of 94.32 million. As the national capital region of China, the Jing-Jin-Ji is the biggest urbanized megalopolis region in northern China and consists of Beijing and Tianjin, two dominant cities and 13 smaller cities in the Hebei Province. As one of the most polluted areas in China, this region had the mean $PM_{2.5}$ of 80 μg/m$^3$ (vs. 52 μg/m$^3$ for mainland China) in 2015.

It has multiple emission sources, and adverse meteorology and terrain—particle matters come from industrial emissions, coal combustion, and motor vehicle exhaust; in winter, heavy coal emission in heating, low wind speed, and high atmospheric stability aggravate concentrating of particle matters; the Mount Taihang-Yanshan surrounding this region may inhibit dispersion of air pollutants from it.

*2.2. Measurement Data*

2.2.1. Satellite-Derived AOD

From the data sharing website of the United States Geological Survey (https://lpdaac.usgs.gov/news/release-of-modis-version-6-maiac-data-products), this study collected daily images (three tiles covering the study region) of MAIAC AOD at 550 nm in 2015 with quality assurance flag data from both Aqua and Terra MODIS, with crossover at approximately 1:30 PM and 10:30 AM local time, respectively. Quality assurance flags provide the retrieval quality of cloud mask, land/water/snow mask and adjacency mask for clouds/snow.

Preprocessing was conducted on the original MAIAC AOD images, including bilinear resampling for projection transformation and filtering of outliers and invalid values according to the valid AOD range (0 to 4) and quality assurance flags. A non-linear GAM was used to fuse Aqua and Terra daily images when one AOD was missing and the other AOD was not missing—non-missing satellite AOD (Aqua or Terra) was used to regress missing AOD (Terra or Aqua), and then both values were averaged into one to increase the size and spatial coverage of training samples (see Supplementary Section S1 for details). Finally, three tiles were mosaicked together into an integrated image covering the study region. As the integral of the aerosol extinction coefficient at all the altitudes within a vertical profile of the atmosphere, satellite AOD is also called column AOD. As one component in the integral of satellite AOD, the ground aerosol extinction coefficient contributes more to high ground-level aerosol loading, and thus is more related to PM$_{2.5}$, compared with satellite (column) AOD. Therefore, a conversion from satellite AOD to ground aerosol extinction coefficient was conducted using planetary boundary layer height (PBLH) and relative humidity (RH) data from the reanalysis data (see Supplementary Section S2.2 for details) using the following formula [69]:

$$k_{AOD,Dry} = \frac{\tau_\alpha}{H_A(1 - RH/100)^{-g}},$$
(1)

where $\tau_\alpha$ is column AOD, *RH* is relative humidity, $H_A$ is the scale height of aerosol, simulated by PBLH [70], $k_{AOD,Dry}$ represents the "dry" aerosol extinction coefficient at surface level that was used in estimation of PM$_{2.5}$, and *g* is an empirical fit coefficient. Here, $(1 - RH/100)^{-g}$ is the ratio of the wet aerosol extinction coefficient obtained as the ambient *RH* to the "dry" one obtained under a relatively dry circumstance under which the PM concentrations are measured in the air sample [4].

2.2.2. AERONET AOD

The AERONET sun photometers collect AOD data at ground level, which are frequently used as the "ground truth" AOD to validate satellite-based AOD. The 2015 2-level AERONET data (version 3) with quality assurance were gathered from two sites (called Beijing and Beijing-CAMS, https://aeronet.gsfc.nasa.gov) located in the Jing-Jin-Ji study region (see Supplementary Figure S2 for their locations). The AERONET AOD data were interpolated to the reference wavelength of 0.55 μm using spectral linear interpolation in the log-log space between the two closest wavelengths within 60 minutes of satellite overpass [71].

2.2.3. Ground Truth PM$_{2.5}$ Measurements

The ground-level hourly PM$_{2.5}$ measurement data (unit: μg/m$^3$) were from 102 monitoring stations of the Ministry of Ecology and Environment of China in the Jing-Jin-Ji study region in 2015 (http://www.pm25.in) (see Supplementary Figure S2 for their spatial locations and 2015 mean).

Daily PM$_{2.5}$ concentration was obtained by averaging hourly measurements using a 75% criterion (measurements covering $\geq$ 18 h/day were used to calculate the daily mean). Using the outer fences [72], 0.1% of the samples were removed as outliers.

### 2.3. Data of the Covariates

MAIAC AOD imputation included the following covariates: coordinates (latitude and longitude) and their derivatives of square and products, elevation, temporal index (Julian day), 1-km gridded surface daily meteorology (air temperature, air pressure, relative humidity, and wind speed) fusing ground measurements and reanalysis data [73,74], daily PBLH, and regional-level daily AOD from the Modern-Era Retrospective analysis for Research and Applications Version 2 (MERRA2) [75]. PM$_{2.5}$ modeling included the following covariates: coordinates (latitude and longitude) and their derivatives of square and products, elevation, daily MAIAC AOD (after imputation), temporal index (Julian day), and the same gridded surface meteorology, PBLH and MERRA2 AOD. For details (i.e., sources, quality and match with the measurement data) about these covariates, please see Supplementary Section S2.

### 2.4. MAIAC AOD for Estimation of Daily PM$_{2.5}$

There is a potential temporal mismatch between MAIAC AOD from the Terra and Aqua sensors and daily PM$_{2.5}$ since satellite overpass time is approximately between 10:30 am and 1:30 pm, and there is NO hourly satellite AOD available to obtain daily averages. But by using the hourly measured AOD of two AERONET sites in the study region, the comparison between the 550 nm AOD mean during the 60-minute window of the overpass time (approximately 9:30 am to 2:30 pm) and the daily 550 nm AOD mean shows a very high significant correlation (0.99, *p*-value < 0.01) between both with a high R$^2$ (0.98). The summary of hourly AOD shows the hourly trend of AOD within a day that suggests the averages between 9:30 am to 2:30 pm approximately approached a daily average of AOD in the study region (average AOD: 0.469 vs. 0.473). The test results showed that the average AOD between 9:30 am to 2:30 pm can be representative of daily AOD with a small difference for the study region.

The daily PM$_{2.5}$ was also compared with the PM$_{2.5}$ mean between 9:30 am and 2:30 pm. The result showed that the correlation between both PM$_{2.5}$ means was 0.95 (*p*-value < 0.01) with a high R$^2$ (0.9), indicating a statistically significant association. Our test also shows the adjusted satellite AOD presented a significant correlation (0.53) with daily PM$_{2.5}$ (*p*-value < 0.01). In statistical terms, the satellite AOD mean between 9:30 am and 2:30 pm, although temporally misaligning with the daily mean, can be used as a valid and reliable predictive variable for estimation of daily PM$_{2.5}$ mean. Similar methods have been used in many existing studies [46,54,59,69,76–78].

## 3. Methods

For both imputation of MAIAC AOD and spatiotemporal estimation of PM$_{2.5}$, the base models of the autoencoder-based residual deep network (Figure 1a) were aggregated in the modeling framework of bagging (Figure 1b). Section 3.1 describes the autoencoder base models, Section 3.2 provides the description of the bagging framework, and Sections 3.3 and 3.4 introduce training and model evaluation respectively.

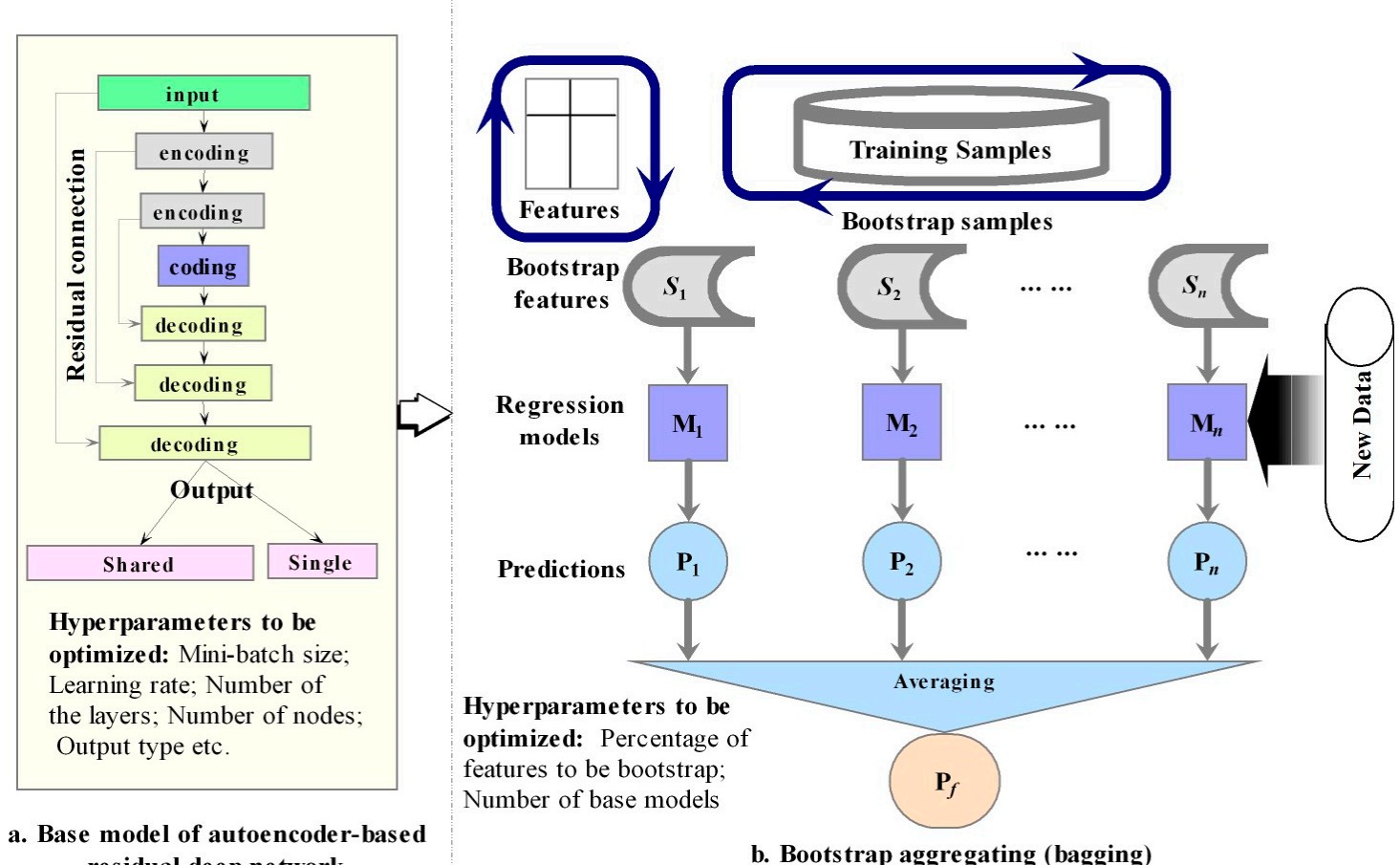

**Figure 1.** Modeling framework.

### 3.1. Autoencoder-based Residual Network

In the base residual network, the autoencoder was introduced to construct the internal network topology (Figures 1a and 2). Autoencoder is a special type of neural network with a symmetrical structure from the encoding to decoding layers and the same input and output variables in an unsupervised or supervised manner [79,80]. It is designed to learn efficient data compression or latent representation coding [81,82] by the middle latent layer (Figure 2), which can improve learning efficiency [83]. Supplementary Section S3.1 presents the autoencoder's basic formulas.

Residual connections have been effectively leveraged to improve learning efficiency in CNN [63,84], and multilayer perceptron [85,86]. Here, residual connections were introduced into the base models of autoencoder. A residual connection needs its starting and ending layers to have the same number of nodes. The autoencoder provides a natural structure for residual connections to be implemented. Residual connections provide the shortcuts from the encoding layer to the decoding layer, and these shortcuts boost the efficient backpropagation of errors from the deep decoding layers to the shallow encoding layers in a parallel way in the learning of the parameters [84,87]. Thus, the introduction of residual connections can effectively solve the potential issues of saturation of gradients and degradation of accuracy in the traditional neural network, as demonstrated in the fusion of the meteorological parameters [85,86]. For technical details, please see Supplementary Section S3.2.

Thus, the base model, autoencoder-based residual deep network consists of the input layer, the internal core of autoencoder, the output layer and residual connections (Figure 2). Residual connections are implemented using the full shortcuts from each encoding layer to its corresponding decoding layer in the autoencoder core, and they can efficiently reduce the issue of gradient vanishing in the neural network of deep layers and improve its performance. The numbers of nodes in the encoding layers in the autoencoder core present a declining trend so as to extract effective latent representations for the input data. The base residual neural network was introduced and successfully applied previously [76,87].

The base models (including an optimal number of layers and the number of nodes for each layer) of residual deep autoencoder for imputation of satellite AOD and spatiotemporal estimation of $PM_{2.5}$ are presented in Figures 2a and 2b respectively. In a typical autoencoder-based residual deep network in Figure 2a,b, the network consists of the input layer, the autoencoder core, and the output layer; the autoencoder core has three parts, i.e., multiple encoding layers, a latent representation layer, and the decoding layers corresponding to the encoding ones; each layer has a number of nodes that represent random variables in this layer; by the weighted summation (the parameters to be learned: weights and bias), each node is linked partially or fully to the nodes of the previous layer and those of the next layer if any. The activation function or/and batch normalization may be added after each layer or before the output layer to improve learning efficiency. As indicated by the dashed lines in Figure 2, residual connections were implemented as the shortcuts between the encoding layers and the corresponding decoding layers.

For regression of a continuous variable, such as MAIAC AOD or $PM_{2.5}$, the autoencoder has been adapted to match the output of the target variables. Based on sensitivity analysis, two different loss functions were defined for the MAIAC AOD imputation and $PM_{2.5}$ estimation:

(1) For the pixel-level MAIAC AOD imputation, there is a large sample size that makes available massive samples for training the outputs of multiple variables (input covariates and the target variable of AOD). Such a multivariable output ensures a better sharing of the parameters between the input covariates and the target variable [88,89] than that of the sole output to reduce over-fitting (Figure 2a). The loss function is defined as follows:

$$L(\theta_{\mathbf{W},\mathbf{b}}) = \frac{1}{N}\Big[\ell_y(y, f_{\theta_{\mathbf{w},\mathbf{b}}}^{(y)}(\mathbf{x})) + \ell_{\mathbf{X}}(\mathbf{x}, f_{\theta_{\mathbf{w},\mathbf{b}}}^{(\mathbf{x})}(\mathbf{x}))\Big] + \Omega(\theta_{\mathbf{w},\mathbf{b}}) \tag{2}$$

where $L(\theta_{\mathbf{W},\mathbf{b}})$ is the total loss, $\mathbf{x}$ is the matrix of covariates, $N$ is the sample size, $y$ is the target variable (MAIAC AOD) to be estimated, $f_{\theta_{\mathbf{w},\mathbf{b}}}^{(y)}(\mathbf{x})$ denotes the model's estimates (output) for $y$ based on the

parameters, $\theta_{\mathbf{W},\mathbf{b}}$, $\ell_y(y, f^{(y)}_{\theta_{\mathbf{w},\mathbf{b}}}(\mathbf{x}))$ denotes the loss of the target variable, $y$, $\ell_{\mathbf{x}}(\mathbf{x}, f^{\mathbf{x}}_{\theta_{\mathbf{w},\mathbf{b}}}(\mathbf{x}))$ is the loss of the model's input $\mathbf{x}$, and $\Omega(\theta_{\mathbf{w},\mathbf{b}})$ denotes the regularizer for $\theta_{\mathbf{w},\mathbf{b}}$, In Equation (2), the extra item, $\ell_{\mathbf{x}}(\mathbf{x}, f^{\mathbf{x}}_{\theta_{\mathbf{w},\mathbf{b}}}(\mathbf{x}))$ works similar to a regularizer for the estimation of $y$.

(2) For the PM$_{2.5}$ spatiotemporal estimation, there is a much smaller sample size than that of the MAIAC AOD, which may result in the trained model with the multivariable output developing a high bias in training. Thus, for PM$_{2.5}$, only a target variable is output with a regularizer (Figure 2b):

$$L(\theta_{\mathbf{W},\mathbf{b}}) = \frac{1}{N}\left[\ell_y(y, f^{(y)}_{\theta_{\mathbf{w},\mathbf{b}}}(\mathbf{x}))\right] + \Omega(\theta_{\mathbf{w},\mathbf{b}}), \tag{3}$$

where $f_{\theta_{\mathbf{w},\mathbf{b}}}(\mathbf{x})$ represents the estimate of $y$, and $\Omega(\theta_{\mathbf{w},\mathbf{b}})$ denotes the regularizer of elastic net [90].

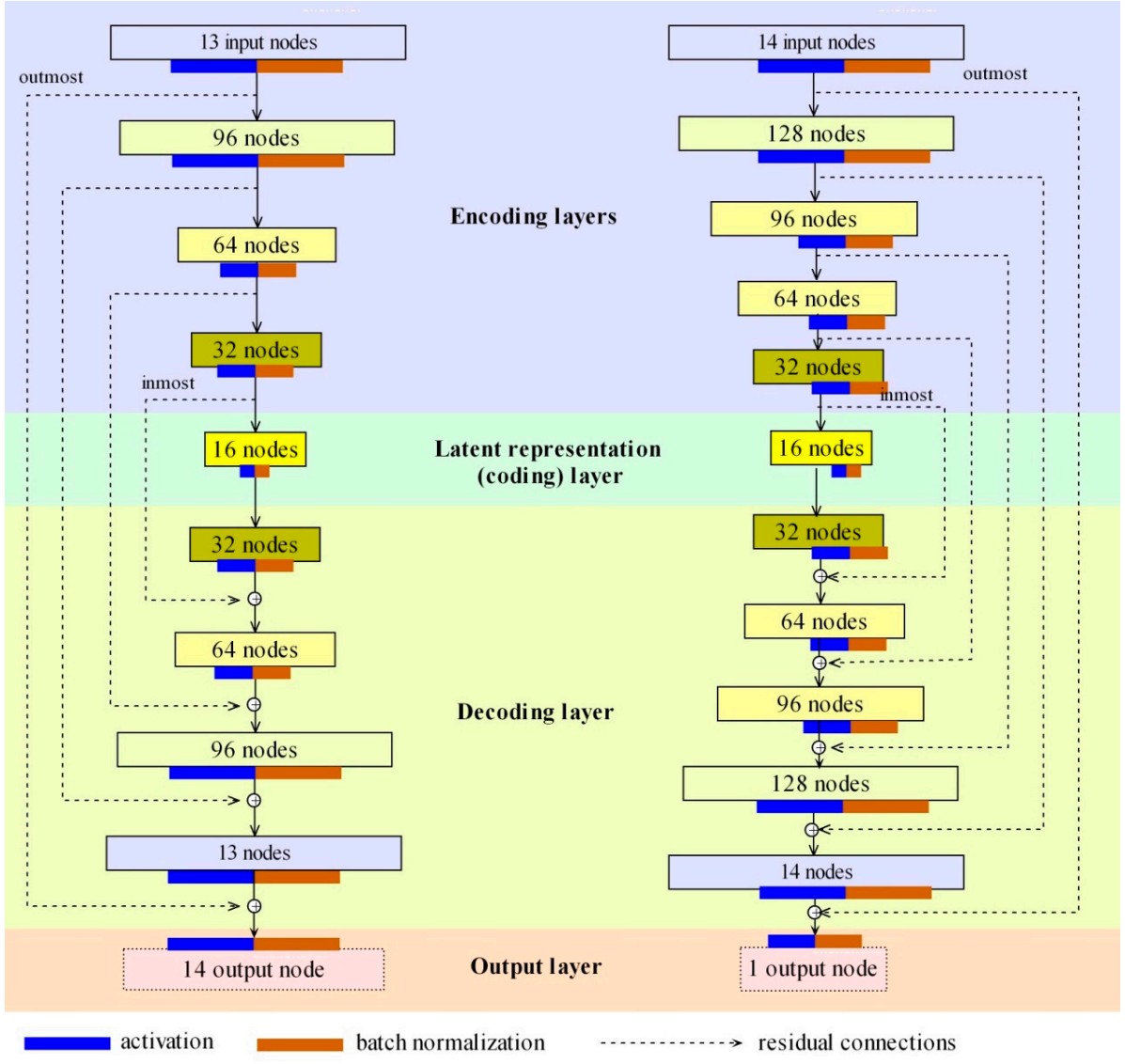

**Figure 2.** Autoencoder-based residual deep networks for imputation of MAIAC (multiangle implementation of atmospheric correction) AOD (aerosol optical depth) (**a**) and spatiotemporal estimation of PM$_{2.5}$ (**b**).

For regression of a continuous variable (y and **x**), the mean square error (MSE) is used for the loss functions. The activation function of rectified linear unit (ReLU) was used in most of hidden layers to maintain efficient backpropagation of the error during training (see Supplementary Section S3.3 for details). For the output layer of a regression of continuous variables, the linear activation function is used. Sensitivity analysis showed that this configuration of the ReLU-linear activation functions effectively prevented the gradients from premature saturation with reliable estimations.

### 3.2. Bagging of Residual Networks

As a model averaging meta-algorithm, bootstrap aggregating (bagging) is designed to improve the stability and performance of single/base machine learning algorithms [91]. It can reduce the variance and over-fitting in the predictions, and extensively applied in regression and classification. The well-known algorithm of random forest is a typical bagging method. In this paper, autoencoder-based residual neural network was used as the base model in bagging.

By the variation in the samples and the features of each bootstrap and the difference in initialization of weights and bias for each training, the trained models can be less related, which can reduce the variance in the predictions by the ensemble models [67]. Assume $n$ models with the errors $\varepsilon_i$ ($I = 1$, $2 \ldots, n$) drawn from a zero mean multivariate normal distribution, $\varepsilon_i \sim MVN\left(0, \begin{pmatrix} v & \ldots & c \\ \ldots & \ldots & \ldots \\ c & \ldots & v \end{pmatrix}\right)$ with the variance $\mathrm{E}\left(\varepsilon_i^2\right) = v$, and covariance $\mathrm{E}(\varepsilon_i \varepsilon_j) = c$. Then, the ensemble prediction determined by averaging is $\frac{1}{n}\sum_i \varepsilon_i$. The expected square error of the ensemble prediction is:

$$\mathrm{E}\left[\left(\frac{1}{n}\sum_i \varepsilon_i\right)^2\right] = \frac{1}{n^2}\mathrm{E}\left[\sum_i\left(\varepsilon_i^2 + \sum_{j\neq i}\varepsilon_i\varepsilon_j\right)\right] = \frac{v}{n} + \frac{(n-1)c}{n} \tag{4}$$

where $c$ represents the covariance between the errors of the models. $c = 0$ indicates no correlation between each model's errors, and the expected squared error of the ensemble averages is $1/n$ of the error variances, i.e., $\frac{1}{n}v$. Furthermore, $c = v$ indicates a perfect correlation between the model's errors, and the expected squared error of the ensemble averages is still $v$, indicating no change.

In the bagging framework (Figure 1b), bootstrapping was conducted for the population samples and the features (covariates) to obtain a set of training samples to train the base model. Stratification by the combined factors of a monthly index and a sample's county index was conducted for each bootstrap to ensure the even distributions of the samples across time and space. For each bootstrap [92], approximately 63.2% of the samples were selected to train a base model, with one-half of the remaining 36.8% samples used as validation samples and the other used for independent tests. For bootstrapping of the features, this method may indicate that only approximately 63.2% of the features would be used to train the models. Such a small number of covariates may result in bias during training. Thus, several important covariates (e.g., latitude, longitude, elevation, temporal index, PBLH, and/or MAIAC AOD) were kept as fixed covariates, and bootstrapping was conducted for the remaining features. Therefore, by bootstrapping the selective covariates, a base model had better generalization than did a complete bootstrapping of the features.

With the bagging of residual networks, the ensemble predictions were obtained to generate the sample mean, standard deviation and standard error. As a metric of uncertainty, the standard deviation can be used with the mean to construct a prediction distribution. Subsequently, the 95% confidence interval of a mean of the bagging predictions can be retrieved through the standard error [93] of each pixel of the grid.

### 3.3. Model Training

For the imputation of MAIAC AOD, the daily pixel-level dataset has a large sample size, ranging from approximately 100,000 to 600,000 pixels (each pixel regarded as a sample). Given a large sample size and the complex interaction between meteorological factors and MAIAC AOD, a daily-level base model was trained with the samples from three continuous days, using the middle day as the target day. In other words, a 3-day window moved continuously over the days from 31 December 2014 to 1 January 2016 so that 365 daily-level models of 2015 were trained to simulate 365 full days of data. The result statistics were also based on all 365 days of estimated results from these daily-level models. If all MAIAC AOD samples for 2015 were combined to train a global model, this would result in a sample size that was too large, and training would be difficult. Daily-level modeling can avoid an intractable sample size and overly demanding computing resources and allows for a temporally varying association between MAIAC AOD and meteorology [54,94]. For the output type, sensitivity analysis showed better performance when the network of covariates shared an output (Figure 2a) than when the non-sharing version was used. The test showed little contribution of regular regularizers of L1 and L2 [95] that are used to reduce over-fitting in machine learning. They did not work well here and thus were not used.

For spatiotemporal estimation of $PM_{2.5}$, given a small sample size (33,119), a single (not daily-level) base model was trained for 2015. For such a sample size, the output of multiple variables might result in bias during training. Sensitivity analysis showed a better generalization, with the $PM_{2.5}$ concentration as the sole output (Figure 2b). For a small training sample size, too many output parameters suppress the target variable's estimation, thus introducing bias during training. The regularizer of the elastic net was used in the model to prevent over-fitting during training [90].

For training, the gradient descent was generally used as the optimizer to find the (sub-) optimal solution for the models. A grid search was leveraged to find an optimal solution for the hyperparameters (see Supplementary Section S4 for details).

For MAIAC AOD imputation, due to the limitation of computing resources, only 10 residual networks were trained for each day by bootstrapping to obtain the ensemble averages, and no standard deviations were generated due to a small number of models. For $PM_{2.5}$, 200 models were trained. Therefore, the ensemble means and their standard deviations were derived through the bagging of residual networks.

### 3.4. Validation and Independent Test

The coefficient of determination ($R^2$), root mean square error (RMSE) and scatter plots with point densities were reported in the training, validation, and testing of the MAIAC AOD and $PM_{2.5}$. To fairly evaluate the proposed approach, GAM, regular feed-forward neural network, and XGBoost were also trained as base models using the same dataset, and the results were compared with those of the proposed method. As a scalable end-to-end tree boosting learning system, XGBoost (https://xgboost.readthedocs.io) (see Supplementary Section S5 for details) is widely used to achieve state-of-the-art results in many domains [65], and Just et al. (2018) [96] used it to correct the measurement errors of MAIAC AOD. Here, as a state-of-the-art method, XGBoost was tested for spatiotemporal estimation of $PM_{2.5}$ and compared with the proposed approach.

For evaluation of the proposed method, the training, validation and independent test $R^2$, RMSE and scatter plots (with sample point density) of the predicted values or residual vs. measured values were reported for the individual base models and ensemble predictions. In addition, for the MAIAC AOD, 365 daily-level ensemble models were trained, and the boxplots of $R^2$ and RMSE were reported with the statistics (mean and range). Four typical seasonal days (i.e., 20 April 2015 for spring, 20 August 2015 for summer, 20 October 2015 for autumn and 20 December 2015 for winter) from 2015 were selected to present the results. Each of the four days selected represented a typical pattern of $PM_{2.5}$ concentration within that day's season (e.g., low in summer and high in winter). For spatiotemporal estimation of $PM_{2.5}$, the ensemble test results were summarized and reported for the whole dataset.

In addition to the independent tests in training, additional independent tests were conducted separately for the imputation of MAIAC AOD and the estimation of $PM_{2.5}$ for rigorous evaluation of the trained models' generalization. For the MAIAC AOD, the measurement data and their interpolated AOD at 550 nm (see Section 2.2.2 for spectral interpolation) were extracted from two AERONET stations (Supplementary Figure S2 for their locations) as the third-party ground truth values to evaluate the imputed satellite AOD. For $PM_{2.5}$, 2015 measured $PM_{2.5}$ data from the monitoring station of the U.S. Embassy (see Supplementary Figure S2 for its location) in Beijing were used as the third-party ground truth values to evaluate the daily grid surfaces of the ensemble-predicted $PM_{2.5}$ for the Jing-Jin-Ji study region.

For practical applications, the normal value ranges were defined for the covariates, AOD and $PM_{2.5}$ to filter possible extreme values (see Supplementary Section S6 for details).

### 3.5. Workflow for Imputation of MAIAC AOD and Estimtion of $PM_{2.5}$

Bagging of residual base models (Figure 2 for the base model; Figure 1 for bagging framework) was developed as the core approach for the imputation of satellite AOD and estimation of $PM_{2.5}$ at high spatiotemporal resolution. Figure 3 presents a workflow graph with the box in yellow indicating the core bagging method. The following seven paragraphs briefly describe specific steps of this workflow.

(1) Collection of satellite AOD data (i.e., MAIAC AOD for this study). This involved matching of the image tiles in time and space with the study region, and data downloading.

(2) Pre-processing of satellite data. This involved re-projection of the satellite images to the target coordinate system, filtering of invalid and noisy AOD, possible mosaic, cropping and masking of image tiles for the study region, and fusion of images from different sources (i.e., Aqua and Terra sensors, see Supplementary Section S1 for details).

(3) Collection and pre-processing of the covariates. The covariates included meteorological factors, MERRA2 data (e.g., PBLH and coarse-resolution AOD), coordinates, and elevation etc. Pre-processing involved the removal of noisy data, re-projection and re-sampling of the data from different sources, and the fusion of meteorological data. The method of the residual deep network was used for the interpolation of meteorological data. For details, please refer to Supplementary Section S2 and [85,86].

(4) Imputation of missing satellite AOD. The core method proposed was used for imputation. This step involved training, validation, and testing of the daily-level imputation models (Figure 2a), and bagging (Figure 3) of multiple outputs. A grid search was conducted to retrieve optimal hyper-parameters for imputation.

(5) The fusion of available and imputed satellite AOD. This involved the mosaic of both AOD, validation of the results, and check of the output to ensure the justification (e.g., a reasonable transition between available and imputed AOD).

(6) Estimation of spatiotemporal $PM_{2.5}$. The covariates dataset from (3) and the satellite AOD with the complete spatiotemporal coverage from (5) were used as the input of explanatory variables. Optimization of the base models (Figure 2b) and bagging was conducted by grid search in training, validation, and testing. Ensemble averaging over the outputs from multiple models was made to get the final mean and standard deviation of $PM_{2.5}$.

(7) Grid output of ensemble predictions and standard deviation (as an uncertainty indicator) of $PM_{2.5}$ at high spatiotemporal resolution.

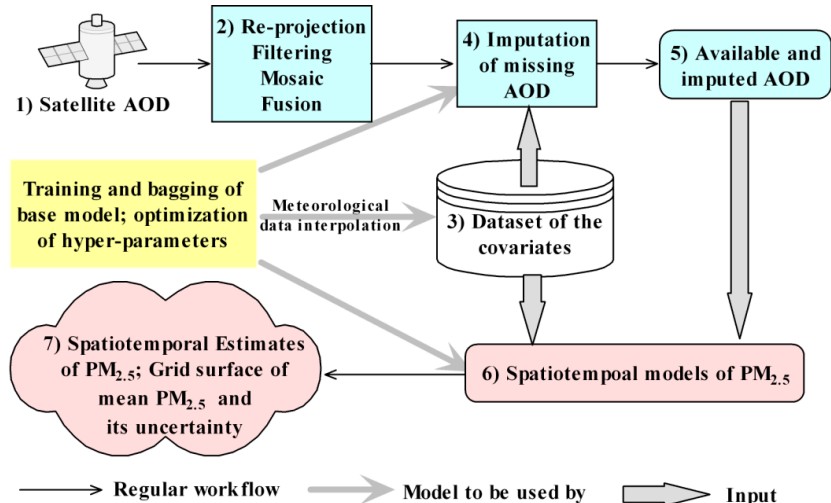

**Figure 3.** A workflow graph for imputation of MAIAC AOD and estimation of PM$_{2.5}$.

## 4. Results

### 4.1. Data Summary

In total, 33,119 valid daily PM$_{2.5}$ measurements were obtained from 102 monitoring stations in 2015. The average concentration was 80.20 μg/m$^3$, higher in the south than that in the north (89.07 μg/m$^3$ vs. 65.40 μg/m$^3$, respectively) (Supplementary Figure S2) and more than twice in winter (January, February, and December) than that in summer (June to August) (121.91 μg/m$^3$ vs. 57.16 μg/m$^3$, respectively). Statistics of the covariates (the mean, standard deviation, and Pearson's correlations with PM$_{2.5}$ and MAIAC AOD) for the whole year and by season are shown in Supplementary Table S1. The mean MAIAC AOD was 0.38, higher in the south than that in the north (Figure 4a). The seasonal trend of summer vs. winter (0.55 vs. 0.43, Supplementary Table S1) AOD was opposite to the seasonal trend of PM$_{2.5}$ (57.16 vs. 121.91μg/m$^3$, Supplementary Figure S2). After the PBLH-RH conversion of column MAIAC AOD to ground aerosol extinction coefficient, the correlation between AOD and PM$_{2.5}$ improved from 0.39 to 0.53.

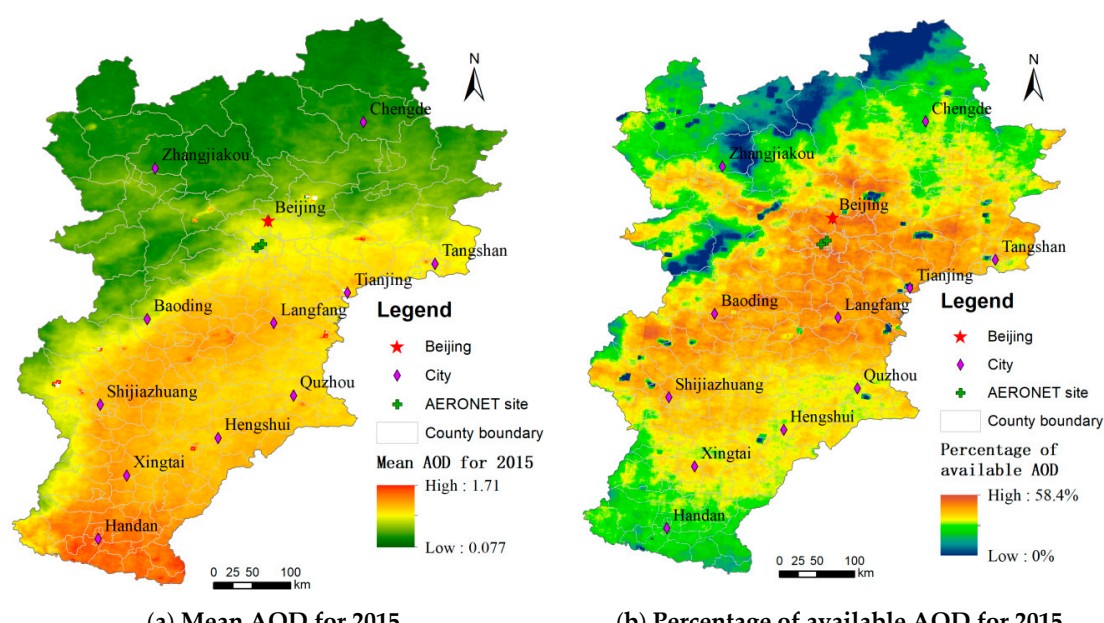

(**a**) **Mean AOD for 2015**　　　　(**b**) **Percentage of available AOD for 2015**

**Figure 4.** Statistics ((**a**) mean AOD; (**b**) percentage of available AOD) for 2015 MAIAC AOD of the study region.

On average, approximately 55% of AOD data were missing for 1-km resolution AOD pixels over 365 days in 2015 (see Figure 4b for spatial distribution), with a higher missing rate in summer than in winter (69% vs. 50%, respectively) (see Supplementary Figure S3b and d for spatial distribution of their availability).

### 4.2. Daily-Level Imputation of MAIAC AOD

The grid search results showed the following optimal options in model performance: a mini-batch size of 512, ReLU as activation functions for hidden layers, and linear unit for the output layer.

Overall, the residual deep network achieved higher mean training and test $R^2$ values (0.96 for both) and mean lower RMSE values (0.057) than those of the regular feed-forward neural network (mean $R^2$: 0.92; mean RMSE: 0.063) and non-linear GAM ($R^2$: 0.86; RMSE: 0.089) (Table 1). Here, training $R^2$ or RMSE was for about 2/3 of the samples used to train the model, while test $R^2$ or RMSE was for about 1/3 of the samples not used in training. The test metrics ($R^2$ and RMSE) reflected the practical performance and generalization for a trained model better than the training metrics. The small difference in $R^2$ and RMSE between the training samples and the test ones indicates no or small over-fitting for the trained model.

The residual deep network improved $R^2$ by 4% compared to the feed-forward neural network and by 10% compared to the GAM. The boxplots for the 365 daily $R^2$ and RMSE values are shown for three models in Figure 5, in which the residual deep network outperforms the other two models.

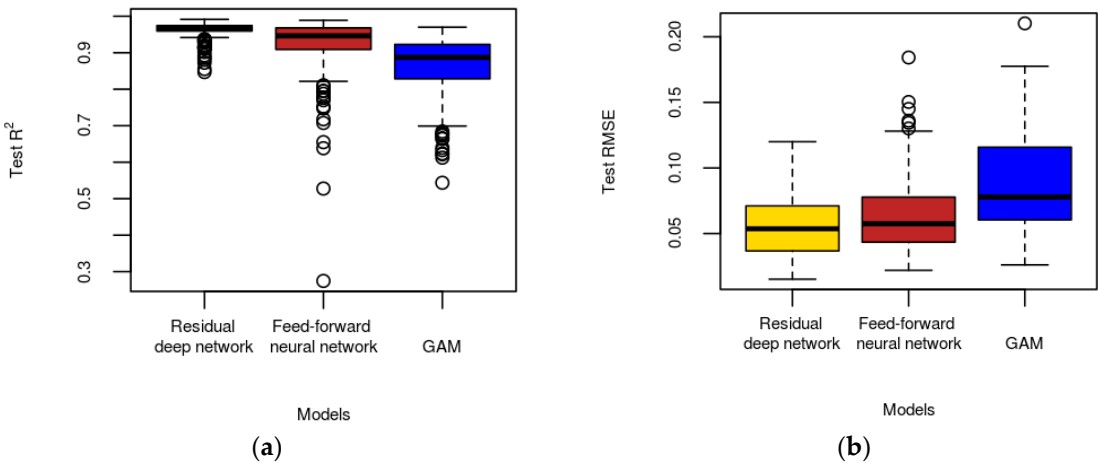

**Figure 5.** Boxplot of $R^2$ (**a**) and RMSE (**b**) for three models, i.e., residual deep network, feed-forward neural network and GAM (generalized additive model) in imputations of daily-level MAIAC AOD (yellow: high performance; orange: moderate performance; blue: low performance).

The AOD imputation results over four typical seasonal days are shown in Table 1 and Figures 6 and 7. Autoencoder-based residual deep networks consistently had higher test $R^2$ ($\geq$0.87) and lower RMSE values than those of the feed-forward neural work, with the largest improvement in winter (test $R^2$: an increase of 8–14%; test RMSE: a decrease of 0.02–0.031). In addition, the loss in the learning curves of the residual networks for the four typical seasonal days showed fast convergence with lower loss than those of the regular feed-forward neural networks (Supplementary Figure S4). Supplementary Figures S5 and S6 show the scatter plots of the observed vs. predicted MAIAC AOD, and their corresponding residual plots respectively in the test with sample point density, demonstrating that most of the test samples (in yellow to red) of MAIAC AOD were accurately estimated by the models.

Missing MAIAC AOD was imputed for each day in 2015 using the trained residual deep networks. The images of the original AOD with missing values and imputed AOD are presented for the four typical seasonal days mentioned above (Figures 6 and 7 and Supplementary Figure S7).

For each of two typical seasonal dates (a spring day: 20 April 2015 for Figure 6; an autumn day: 20 October 2015 for Figure 7), additional pixels having available observed MAIAC AOD in the southern (for 20 April 2015) or northern (for 20 October 2015) sub-region were removed as missing values. In total, approximately 50% of available MAIAC AOD was used as observed values to validate the MAIAC AOD imputed by the re-trained models. In other words, the remaining 50% of samples of the available pixels were used to re-train the models. The performances (test $R^2$: 0.95–0.96) of the re-trained models (reported in Supplementary Table S2) were similar to those (test $R^2$: 0.96–0.97) of the models trained using all the samples and the predicted AOD for the additional missing values show reasonable transition between available and imputed MAIAC AOD based on their original observed values to a certain degree. The models trained using different samples due to AOD availability had the differences in the imputed values at some locations, as shown in b vs. d in Figures 6 and 7. Such differences were substantial for the models trained using a small size of available AOD samples (e.g., for 20 October 2015), compared with those trained using a big size of samples (e.g., for 20 April 2015). Thus, for a high missing proportion of AOD samples, the limited samples may be insufficient for a valid evaluation, particularly for the spatial pattern of the massively imputed AOD. Here, the correlation between the imputed AOD and co-located $PM_{2.5}$ measurements was used indirectly to evaluate the imputed AOD. The resultant correlation (0.61) shows that the imputed AOD moderately captured spatial variability of aerosol mass for 20 April 2015.

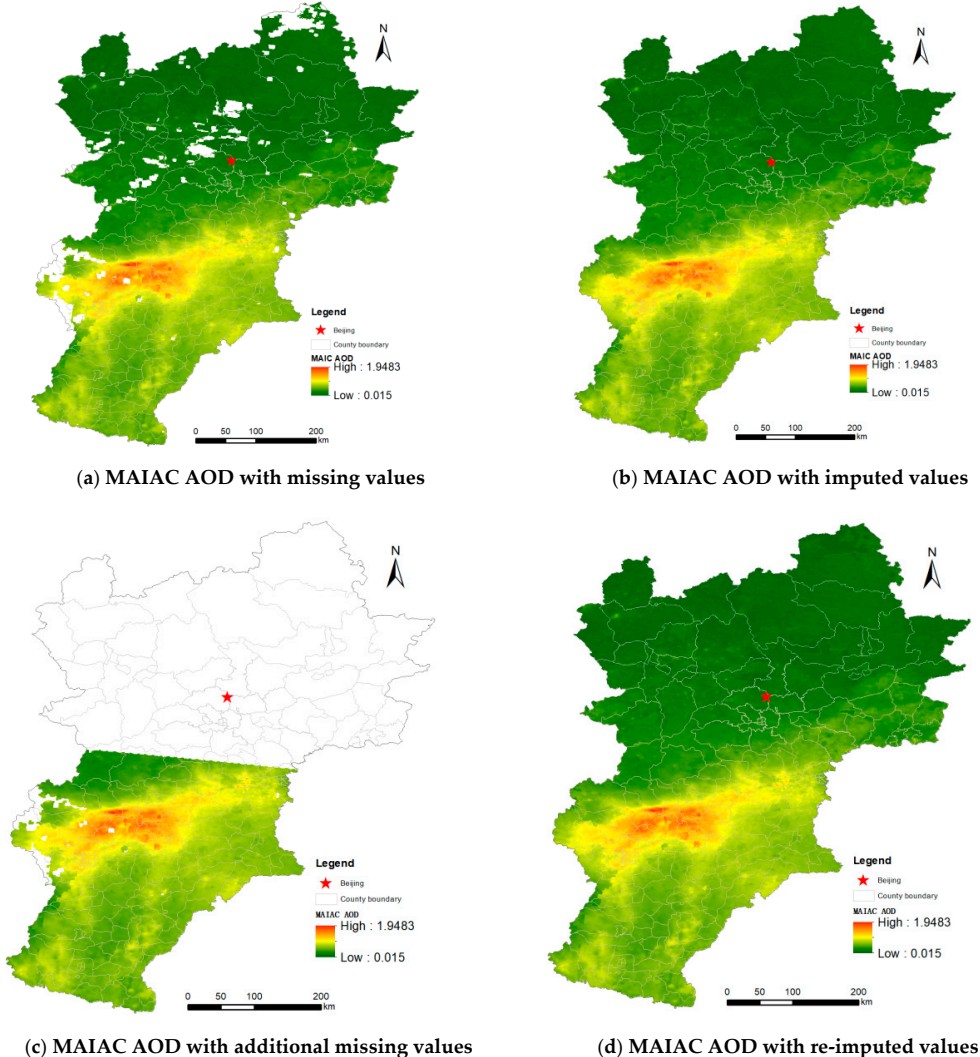

(**a**) **MAIAC AOD with missing values**

(**b**) **MAIAC AOD with imputed values**

(**c**) **MAIAC AOD with additional missing values**

(**d**) **MAIAC AOD with re-imputed values**

**Figure 6.** Original (**a**) and imputed MAIAC AOD (**b**), MAIAC AOD with additional missing values (**c**) and MAIAC AOD of re-imputed values (**d**) for a spring day (20 April 2015).

Figures 6a and 7a present the original MAIAC AOD of two days (representative with minimum missing values (spring) and substantial missing values (autumn)), with a low (a) and a high (c) missing proportion. The imputed results showed reasonable patterns of spatial variation between the available and imputed AOD for the four typical seasonal days, illustrating reliable imputation results.

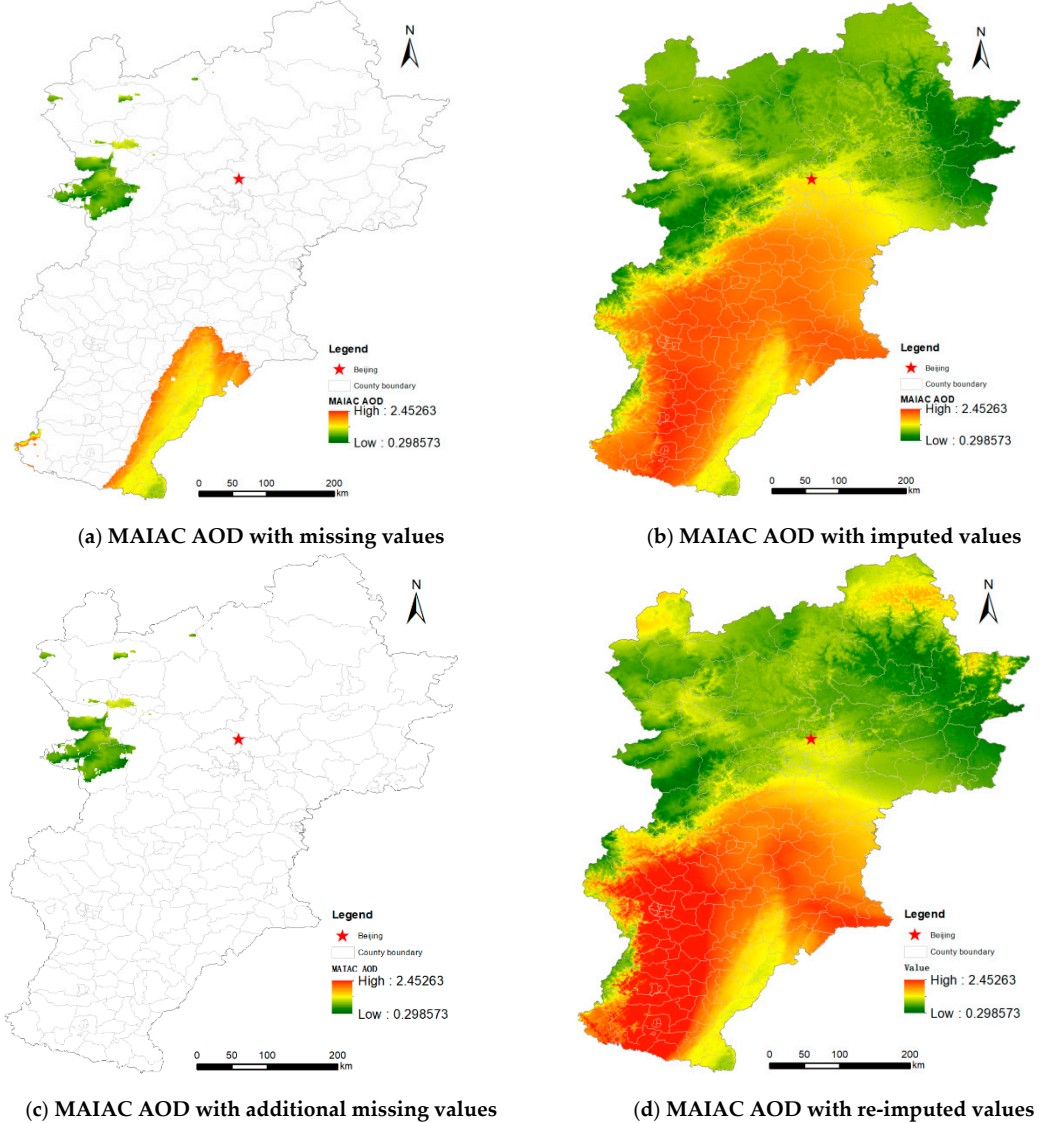

**Figure 7.** Original (**a**) and imputed MAIAC AOD (**b**), MAIAC AOD with additional missing values (**c**) and MAIAC AOD of re-imputed values (**d**) for an autumn day (20 October 2015).

**Table 1.** Comparison of GAM, feed-forward neural network and residual deep network for daily-level MAIAC AOD imputation.

| Date | Model | Training $R^2$ | Training RMSE | Test $R^2$ | Test RMSE |
|---|---|---|---|---|---|
| Averages for all days (Range[a]) | Residual Deep Network | 0.96 (0.85 to 0.99) | 0.058 (0.015 to 0.12) | 0.96 (0.85 to 0.99) | 0.057 (0.011 to 0.15) |
| | Feed-forward Neural Network | 0.90 (−0.2 to 0.91) | 0.065 (0.022 to 0.21) | 0.92 (0.27 to 0.98) | 0.063 (0.018 to 0.22) |
| | GAM | 0.86 (0.55 to 0.97) | 0.089 (0.026 to 0.21) | 0.86 (0.54 to 0.97) | 0.089 (0.021 to 0.26) |
| 04/20/2015 | Residual Deep Network | 0.97 | 0.072 | 0.97 | 0.073 |
| | Feed-forward Neural Network | 0.95 | 0.078 | 0.95 | 0.078 |
| | GAM | 0.90 | 0.089 | 0.90 | 0.088 |
| 07/20/2015 | Residual Deep Network | 0.97 | 0.13 | 0.96 | 0.12 |
| | Feed-forward Neural Network | 0.92 | 0.15 | 0.93 | 0.14 |
| | GAM | 0.91 | 0.15 | 0.91 | 0.15 |
| 10/20/2015 | Residual Deep Network | 0.96 | 0.14 | 0.96 | 0.14 |
| | Feed-forward Neural Network | 0.92 | 0.13 | 0.92 | 0.13 |
| | GAM | 0.88 | 0.17 | 0.88 | 0.17 |
| 12/01/2015 | Residual Deep Network | 0.93 | 0.061 | 0.87 | 0.064 |
| | Feed-forward Neural Network | 0.74 | 0.086 | 0.79 | 0.084 |
| | GAM | 0.74 | 0.094 | 0.73 | 0.095 |

Note: Range[a]: The interval between the minimum value and the maximum one among the performance metrics.

### 4.3. Spatiotemporal Estimation of PM$_{2.5}$

The residual deep network for PM$_{2.5}$ prediction (Figure 2b) achieved a higher mean test R$^2$ (0.86) and lower mean RMSE (26.40 µg/m$^3$) than did the feed-forward neural network and GAM (R$^2$: 0.48–0.78; RMSE: 34.21–50.84 µg/m$^3$) (Table 2). The loss learning curve (Supplementary Figure S8) shows better convergence with a lower loss for the residual deep network than for the regular feed-forward neural network. Compared with XGBoost, the individual base residual network achieved similar training performance with a slightly lower test R$^2$ (approximately 3%). However, bagging of the residual networks achieved similar test R$^2$ (0.90) and RMSE (22.4 µg/m$^3$) for the ensemble predictions as those from XGBoost. The ensemble predictions of the residual networks improved more over the base network (4%) than those of XGBoost and GAM (1%). Scatter plots (Figure 8) of the predicted (a) and residual (b) PM$_{2.5}$ vs. observed PM$_{2.5}$ with sample point density showed an accurate match (test R$^2$: 0.90) between the ensemble predictions and the observed PM$_{2.5}$.

**Table 2.** Comparison of GAM, feed-forward neural network and residual deep network for PM$_{2.5}$ spatiotemporal estimation.

| Model | Type | Training R$^2$ | Training RMSE | Test R$^2$ | Test RMSE | Bagging Test R$^2$ | Bagging Test RMSE |
|---|---|---|---|---|---|---|---|
| Residual Deep Network | Mean/Total[a] | 0.89 | 23.08 | 0.86 | 26.32 | 0.90 | 22.40 |
| | Range[b] | 0.86–0.92 | 21.76–26.71 | 0.82–0.89 | 23.17–29.95 | - | - |
| XGBoost | Mean/Total | 0.88 | 24.07 | 0.89 | 23.06 | 0.90 | 22.41 |
| | Range | 0.86–0.89 | 22.14–27.25 | 0.85–0.90 | 21.73–26.68 | - | - |
| Feed-forward Neural Network | Mean/Total | 0.82 | 31.61 | 0.78 | 34.21 | 0.84 | 25.48 |
| | Range | 0.77–0.85 | 26.34–35.56 | 0.73–0.82 | 29.83–38.35 | - | - |
| GAM | Mean/Total | 0.48 | 50.66 | 0.48 | 50.84 | 0.49 | 49.82 |
| | Range | 0.47–0.49 | 48.97–52.98 | 0.46–0.50 | 48.97–52.98 | - | - |

Note: Mean/Total[a]: mean of 200 models for training and testing; total of ensemble predictions for each test sample by bagging of the models; Range[b]: indication of minimum and maximum of the metrics of 200 models, no value available for bagging test.

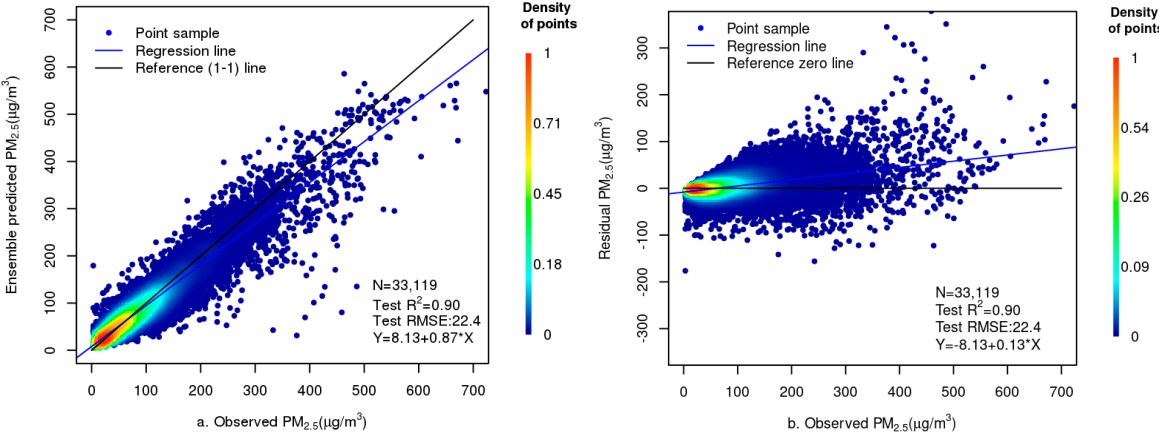

**Figure 8.** Scatter plots of observed (**a**) and residuals (**b**) vs. observed PM$_{2.5}$ (µg/m$^3$) in the test data of 2015.

The predicted PM$_{2.5}$ grid surfaces with standard deviations of the four typical seasonal days are presented in Figure 9 (spring and autumn) and Supplementary Figure S9 (summer and winter). The seasonal and spatial distributions of PM$_{2.5}$ on these days are similar to the yearly average results as discussed earlier. Standard deviations from ensemble PM$_{2.5}$ predictions are shown in Figure 9b,d and Supplementary Figure S9b,d. Supplementary Figure S10 shows 95% confidence interval results, which can be used in health outcome analysis as a measure of uncertainty in PM$_{2.5}$ predictions.

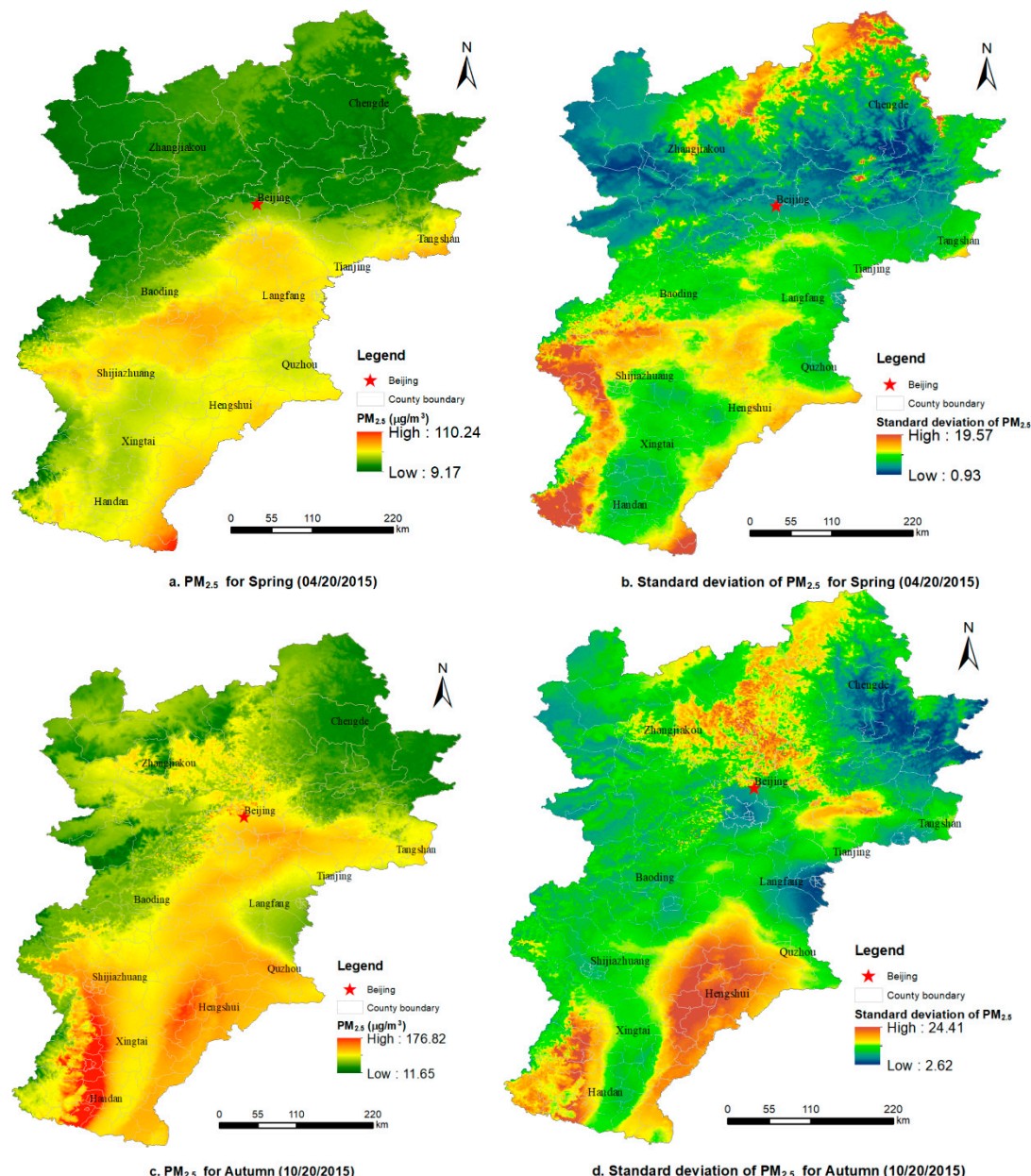

**Figure 9.** Grid surfaces of predicted PM$_{2.5}$ (μg/m$^3$) ((**a**,**c**)) and its standard deviation ((**b**,**d**)) for two seasonal days ((**a**,**b**): spring; (**c**,**d**): autumn) in 2015.

Ensemble-predicted PM$_{2.5}$ grid surfaces obtained by XGBoost are presented in Supplementary Figure S11. Despite similar spatial distributions on a large scale, PM$_{2.5}$ predictions from XGBoost showed problematic results (i.e., abrupt and unrealistic variations) at certain locations on a local scale.

### 4.4. Additional Independent Test

In addition to the independent test results reported in Tables 1 and 2, additional independent tests based on the ground truth AOD from two AERONET sites and the ground truth PM$_{2.5}$ measurements from a monitoring station at the U.S. Embassy in Beijing showed consistent agreement between the ensemble predictions and observed values for both MAIAC AOD (total correlation: 0.93 with R$^2$ of 0.82 and RMSE of 0.212) and PM$_{2.5}$ predictions (total correlation: 0.99 with R$^2$ of 0.97 and RMSE of 12.23 μg/m$^3$) (Table 3). Scatter plots between predicted and observed values are presented in Supplementary Figure S12, illustrating reliable model performance.

In addition, the original available MAIAC AOD against AERONET AOD shows a slightly higher correlation and $R^2$, and slightly lower RMSE than the complete AOD (including available and imputed values) (Table 3). The difference in the performance is very small, and the validation shows the reliability of the proposed approach.

**Table 3.** Validation of residual deep network with AERONENT AOD (MAIAC AOD) and U.S. Embassy monitoring station ($PM_{2.5}$).

| Type | Site | #Samples | Correlation | $R^2$ | RMSE |
|---|---|---|---|---|---|
| | Beijing (Available and imputed AOD) | 231 | 0.92 | 0.80 | 0.202 |
| | Beijing (Available AOD) | 167 | 0.94 | 0.83 | 0.184 |
| MAIAC AOD | Beijing-CAMS (Available and imputed AOD) | 274 | 0.93 | 0.82 | 0.220 |
| | Beijing-CAMS (Available AOD) | 190 | 0.95 | 0.84 | 0.191 |
| | All | 505 | 0.93 | 0.82 | 0.212 |
| $PM_{2.5}$ | U.S. Embassy station | 365 | 0.99 | 0.97 | 13.23 μg/m$^3$ |

## 5. Discussion

### 5.1. Strengths of Bagging of Residual Networks

This paper presents a novel approach, i.e., bagging of residual networks for robust imputation of MAIAC AOD and estimation of $PM_{2.5}$ at a high spatiotemporal resolution. In this approach, autoencoder was used as the base to implement residual connections to boost backpropagation of errors in learning and hence reduce gradient saturation and accuracy degradation and improve model performance. Potential model over-fitting was reduced by the parameter sharing in the multivariable output (for MAIAC AOD) or the use of an elastic net (for $PM_{2.5}$). Ensemble learning through bagging of autoencoder-based residual networks further improved model performance. The presented approach could reliably address massive non-random missingness of satellite AOD and hence predict $PM_{2.5}$ at high spatiotemporal resolution and with a complete spatiotemporal scale. For the challenges of massive missing satellite AOD, limited data of $PM_{2.5}$ and publicly available covariates, and local variation of concentration probably caused by more traffic emissions than previously, compared with other existing methods, this proposed approach achieved $PM_{2.5}$ prediction with complete spatial coverage, a high spatiotemporal resolution, and cutting-edge performance. Particularly, in winter, complex meteorological conditions (more clouds and snow) caused the difficulty in training the GAM and feed-forward neural network for AOD imputation than in other seasons. The residual deep network achieved much better performance than the other two models in winter, illustrating its robustness under challenging conditions. The accurate spatiotemporal estimation of $PM_{2.5}$ has an important implication for the reduction of the bias in exposure assessment and in turn the epidemiological studies of health effects.

As a globally optimal method, support machine vector has been increasingly used for $PM_{2.5}$ estimation [97–99]. However, the scalability of this method is constrained in studies with a large sample size (e.g., high spatiotemporal resolution AOD over a long period and a large region). In addition, choosing the appropriate hyperparameters and kernel functions requires time-consuming feature engineering operations and expert knowledge [100]. Comparatively, the bagging of residual deep network approach is easier to implement, requiring fewer manual operations, and achieving reliable model performance with publicly available limited covariates.

Furthermore, bagging improved test $R^2$ by 4–6% over the individual base models for residual networks and regular feed-forward neural networks; the improvement was minimal (1%) for GAM and XGBoost (Table 2). Since XGBoost is an ensemble learner based on multiple decision trees, further bagging of multiple XGBoost has limited contribution to model performance. GAM uses a back-fitting algorithm to fit parameters, which results in small variation in model parameters and hence small differences between trained models and subsequently little improvement in bagging predictions. Comparatively, as a non-ensemble learner, the residual deep network has different initializations of the

parameters and local optimal solutions attained by gradient descent, which makes the trained models from bootstrapping less related, thus more improvement from bagging than GAM and XGBoost.

With small changes (e.g., input and output) and tuning by regular grid search, the modeling framework (Figures 1 and 2), and the hyperparameters used (Supplementary Section S5) may be applied to obtain an optimal solution for spatiotemporal estimation of similar air or other environmental pollutant variables including satellite AOD and PM$_{2.5}$. The solution may not be globally optimal but is often better than traditional methods and applicable to many practical settings [67]. The open-source package of Python (https://pypi.org/project/baggingrnet) for the proposed approach was published, which provides available functions and specific examples to illustrate the applications of the proposed approach.

For the imputation of satellite AOD and estimation of PM$_{2.5}$, the network structure (Figure 2) and loss functions (Equations (2) and (3)) proposed in this study can be used to train a locally optimal model that is often useful sufficiently for practical applications [67]. With this as a benchmark model, and depending upon the time and resources available, grid search may be conducted limitedly or extensively to find optimal values for hyperparameters to have a better performance. For the implementation of the proposed approach, initialization and specific values of several important hyperparameters are given and primary optimizations are briefly described in Supplementary Section S5—the workflow (Figure 3) with specific steps is given.

### 5.2. Spatiotemporal Variability of Predicted PM$_{2.5}$

As one of the world's most polluted areas, the Jing-Jin-Ji study region has a high annual PM$_{2.5}$ concentration, far exceeding the World Health Organization PM$_{2.5}$ annual standard of 10 μg/m$^3$ [101]. The major sources of PM$_{2.5}$ include coal combustion in industrial and residential areas and emissions from automobile exhaust [21]. Due to dense industrial plants and population, the middle-southern urban areas have higher PM$_{2.5}$ concentrations than do the northern rural and remote areas. Additionally, due to the increase in coal consumption for heating, winter has much higher PM$_{2.5}$ concentrations than those of other seasons. Autumn has higher PM$_{2.5}$ concentrations than those of spring and summer likely due to the influence of meteorology (lower wind speed). Daily grid surfaces of predicted PM$_{2.5}$ (Figure 7 and Supplementary Figure S9) are reasonable in seasonal and spatial patterns. To examine how spatiotemporal autocorrelation was captured by the proposed approach, daily Moran's I [102] was calculated for the residuals. The results showed *p*-value > 0.05 for 150 days, indicating complete spatially random predictions; for the other 215 days, Moran's I was small (mean: 0.09), indicating that dominant spatial autocorrelation was captured by the trained models through the input of proxy variables (e.g., coordinates, Julian day and spatiotemporally varying AOD).

Furthermore, the ensemble predictions of PM$_{2.5}$ by bagging of residual networks presented better spatial natural variation than did XGBoost. The use of decision trees as the core component in XGBoost resulted in the discretization of continuous covariates that, in turn, caused the abrupt spatial variation of predicted PM$_{2.5}$ at some locations (Supplementary Figure S11). Comparatively, the residual deep network maintained the full range of continuous covariates (no discretization) in the trained models, which could avoid potential inaccuracy resulting from discretization in decision trees.

### 5.3. Uncertainty in Predicted PM$_{2.5}$

The spatiotemporal variability of PM$_{2.5}$ concentration is affected by multiple factors (e.g., meteorology, emission sources, and land-use) and their interactions. Uncertainty in sampling and input variables directly leads to uncertainty in predicted PM$_{2.5}$ concentrations. Intuitively, high uncertainty in the input data can result in high standard deviation, thus uncertainty in the predictions, indicating a wide confidence interval. Such uncertainty, if well-characterized, can be used to simulate exposure prediction errors. One strength of bagging is to generate the standard deviation of a predicted mean as a measure of uncertainty, which can further be used in health outcome analysis [103–105].

Hyperparameters (e.g., mini-batch size, activation function (sigmoid, tanh, ReLU and linear) and output type (multivariable or sole-variable)) showed important influences on the standard deviation of predicted $PM_{2.5}$. Hyperparameters had a significant effect on the reduction of standard deviation. A search for optimal solutions of hyperparameters is necessary to improve model performance and computing efficiency and decrease potential bias. Additionally, for this study, 200 models were trained due to limitations of computing resources (the computing configuration of an Intel (R) Xeon(R) with 8 CPUs (E5530, 2.4G) and 32 G memory), but with higher capability computing systems, more models can be trained that can possibly further reduce standard error in prediction.

The correlations analysis between the standard deviation of the predicted $PM_{2.5}$ and the influential factors (Supplementary Table S3) showed more contributions of MAIAC AOD, longitude, and meteorology (air temperature, air pressure, and elevation) to the standard deviation than those of other factors (PBLH, MERRA2 AOD, and latitude). Compared with the other covariates, MAIAC AOD is most associated with the standard deviation of ensemble-predicted $PM_{2.5}$, which was determined by its complex spatiotemporal variation subject to multiple factors including meteorology, emission source, and surface reflectance.

*5.4. Limitation and Prospects*

The first limitation of this study is the uneven spatial distribution of $PM_{2.5}$ training data, which are highly concentrated in dense traffic and urban areas and are underrepresented in remote areas. Here, stratification by county and month was conducted in bootstrapping to ensure an even distribution of the samples across space and time. Because epidemiological studies primarily focus on populated areas rather than rural and remote areas, such bias in exposure estimation is limited when evaluating the health effects of $PM_{2.5}$. The second limitation is the proposed method's applicability for the imputation of MAIAC AOD having a high missing proportion (e.g., >80%). For high missing AOD, the samples of actually high or low AOD may be completely unavailable and thus, the trained models may be biased and tend to be under- or overestimate the actual AOD. In the proposed imputation method, a 3-day window was used to obtain a large sample rather than a day sample, and the added samples temporally lagged before or after the target day can to a certain extent compensate the shortcoming of the samples for a target day having a high proportion of missing AOD. In total, the proposed imputation method achieved reasonable imputed values as shown in statistical performance metrics although having limited over- or underestimations at some pixels. In addition to the validation based on the limited samples, the correlation between the imputed AOD and the $PM_{2.5}$ measurement data may be used to evaluate the spatial variability of the imputed AOD. The third limitation is the lack of two relevant variables, i.e., cloud fraction for AOD, and land-use for $PM_{2.5}$, in the models due to unavailability. Both variables may further improve the models. However, this has a limited influence upon the results as shown in the test and validation. The fourth limitation is the temporal misalignment between satellite AOD and daily $PM_{2.5}$ mean since satellite AOD is available only during the overpass time (approximately between 9:30 am and 2:30 pm) of each day. However, the data samples showed that such misalignment just resulted in a small difference between both AOD means. In statistical terms, satellite AOD was still a valid and reliable predictor for estimation of daily $PM_{2.5}$ given a statistically significant correlation between both.

The proposed approach was used in the spatiotemporal estimation of meteorological parameters in a considerably extensive region, i.e., mainland China with reliable performance [85,86]. In the future, the proposed approach will be used to predict $PM_{2.5}$ concentration at a high spatial (1 km) and temporal (daily) resolution for an extensive spatial (mainland China) and temporal (2000–2016) domain. This will provide useful data for epidemiological studies of $PM_{2.5}$ in China, particularly for the period of 2000–2012 when no national $PM_{2.5}$ monitoring data were available. Furthermore, with $PM_{2.5}$ constituent measurements available, this approach can be used to explore their spatiotemporal trends in mainland China.

## 6. Conclusions

This paper presents an approach of deep learning, i.e., bagging of autoencoder-based residual networks for robust imputation of massive missing satellite AOD and estimation of $PM_{2.5}$ concentrations at a high spatiotemporal resolution. In the proposed approach, the autoencoder was used to extract the latent representation and implement the residual connections to boost training and improve generalization. For a large training sample size with the MAIAC AOD imputation, the output of multiple variables enabled the sharing of parameters to reduce overfitting. In the case study of the Jing-Jin-Ji metropolitan region, this approach achieved higher test $R^2$ (0.96 for MAIAC AOD; 0.90 for $PM_{2.5}$) and lower RMSE (0.06 for MAIAC AOD; 22.3 $\mu g/m^3$ for $PM_{2.5}$), compared with those of the regular feed-forward neural network and GAM. Compared with the state-of-the-art machine learning method, XGBoost, this approach generated better spatial variation for grid surfaces of predicted $PM_{2.5}$. Furthermore, the bagging of residual networks generated the mean and standard deviation for the distribution of a pixel-predicted $PM_{2.5}$. With full spatial coverage by the imputation of missing MAIAC AOD, the high-accuracy estimation of $PM_{2.5}$ at a high spatiotemporal resolution can considerably reduce the bias in exposure estimation for research on the acute, short-term, and chronic health effects of $PM_{2.5}$.

**Supplementary Materials:** The following are available online at http://www.mdpi.com/2072-4292/12/2/264/s1, Section S1: Fusion of daily Aqua and Terra AOD, Section S2: Covariates, Section S3: Autoencoder-based residual network, Section S4: Optimization of hyperparameters, Section S5: XGBoost; Section S6: Processing of extreme values, Figure S1: Distribution of 2014 vs. 2015 $PM_{2.5}$ emission sources in Beijing, Figure S2: Study region with distributions and 2015 annual averages of $PM_{2.5}$ monitoring stations, AOD sites of AERONET, and monitoring stations of the US embassy in Beijing, Figure S3: Statistics for summer (a and b) and winter (c and d) of 2015, Figure S4: Learning curves of test loss of four typical seasonal days of 2015 for MAIAC AOD imputation, Figure S5: Plots of observed vs. imputed daily MAIAC AOD in the test for four typical seasonal days in 2015, Figure S6: Plots of observed vs. residual MAIAC AOD for four typical seasonal days in 2015, Figure S7: Original (a and c) and imputed MAIAC AOD (b and d) of the study region for two seasonal days (a and b: summer; c and d: winter) in 2015, Figure S8: Learning curves of validation loss for $PM_{2.5}$ estimation, Figure S9: Grid surfaces of predicted $PM_{2.5}$ ($\mu g/m^3$) (a and c) and its standard deviation (b and d) for two seasonal days (a and b: spring; c and d: autumn) in 2015, Figure S10: Images for low (a) and high (b) bounds of the 95% confidence interval of predicted $PM_{2.5}$ (winter day of /12/01/2015), Figure S11: Grid surfaces of predicted $PM_{2.5}$ ($\mu g/m^3$) by XGBoost for four seasonal days (a: spring; b: summer; c: autumn; d: winter) in 2015 (representation with spatial discontinuity), Figure S12: Scatterplots between observed vs. predicted values in the independent tests of (a) MAIAC AOD using two AERONET sites and (b) $PM_{2.5}$ using the US embassy monitoring station, Table S1: Descriptive Statistics for the covariates and target variables (2015), Table S2: The performance of the re-trained models for MAIAC AOD, Table S3: Pearson's correlation between uncertainty (standard deviation) of grid surfaces of $PM_{2.5}$ and the covariates.

**Author Contributions:** Lianfa Li was responsible for conceptualization, methodology, software, validation, formal analysis and writing. All authors have read and agreed to the published version of the manuscript.

**Funding:** This work was supported in part by the National Natural Science Foundation of China under Grant 41471376 and in part by the Strategic Priority Research Program of Chinese Academy of Sciences Grant XDA19040501.

**Acknowledgments:** The support of NVIDIA Corporation with the donation of the Titan Xp GPUs used for this research and Ying Fang's support for this study are gratefully acknowledged.

**Conflicts of Interest:** The author declares no conflict of interest.

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
