# Peer review of "A Robust Deep Learning Approach for Spatiotemporal Estimation of Satellite AOD and PM2.5"

_remotesensing, doi:10.3390/rs12020264_

Round 1

Reviewer 1 Report

I congratulate and thank the author for the extensive work done to validate the results and suggest that the manuscript should be published in its current state.

Reviewer 2 Report

The author presents a novel approach to derive aerosol optical depth and associated particulate matter < 2.5 µm from satellite, to be more precise MODIS on Aqua and Terra. A major issue with deriving AOD from Satellite is the obscuration by clouds, the retrievals work only satisfactorily well in clear-sky conditions. Thus, a usual AOD scene from MODIS consists of a large number of missing pixels. The authors presents a method to fill the missing AOD pixels with the help of a novel machine learning method (autoencoder-based residual networks with bagging of the networks) and derive PM2.5 from the AOD fields. Furthermore the author compares the new method to three standard machine learning approaches. The new method works impressively well and even outperforms the other algorithms.

Dear author, to my mind, the work is presented in a clear and concise way. Some parts need a bit more clarification to be more understandable for the reader. I would recommend the article for publication with minor corrections and improvements.

Minor comments:

All pages: the article seems to be in correction mode, I received a copy with many crossed out lines or lines that were added, furthermore the page numbering was somehow totally off, I had several pages numbered with 2 or 4. Probably there is an issue with the style file of Remote Sensing? I am sure this can easily be resolved.

L 135: conduct -> conducted

L 143: It would be very interesting to have an average number of PM 2.5 load for comparison. About 80 µg/m³, as mentioned later, is really remarkable.

L 160: Explain the difference between AOD and ground AOD in more detail.

L 162: Planetary boundary layer height and relative humidity were probably taken from reanalysis? Please mention it at this position to avoid confusion for the reader.

Formula 1: there is probably a 1/100g missing, compared to reference 69.

Ll 204 – 210: Please explain the autoencoder network in more detail, since it is the main method in your article, it needs to be made more comprehensive.

Section 3.2: Same comment as before, please explain the concept of bagging in more detail.

Figure 2 definitely needs an explanation, simply mentioning the number of nodes does not contain information for the reader.

L 389: a space is needed after 55%.

Figure 4: Legends are very small “Beijing” is almost unreadable when printed.

L 263: do you mean “count id”?

Figure 6 c) and Figure 7 c): There seem to be many more pixels missing than 20% compared to a) (as mentioned in the caption). Could you please explain?

L642 or section 5.3: Do you have a rough estimate on how dependent the methods are on uncertainties in training data? Do you think the methods could work too in regions with less aerosol load?

Reviewer 3 Report

Please see attached pdf.

Round 2

Reviewer 3 Report

I agreed the author's amendments in this time, and responses fully addressed my concerns. I appreciate the author's work.

This manuscript is a resubmission of an earlier submission. The following is a list of the peer review reports and author responses from that submission.

Round 1

Reviewer 1 Report

A Robust Machine Learning Approach for Spatiotemporal Estimation of Satellite AOD and  PM2.5

Peer Review Report
Review of “A Robust Machine Learning Approach for  Spatiotemporal Estimation of Satellite AOD and PM2.5”

By Lianfa Li

This paper represents  an approach of machine learning, i.e., bagging of autoencoder-based  residual networks for robust imputation of massive missing satellite AOD and estimation of PM2.5  concentrations at a high spatiotemporal resolution.

I'm not an expert on machine learning, which is why I will refer to the results of modeling PM2.5 and AOD. The author used several computational methods and presented their comparison (Tab. 1). This is a very important approach to the issue.

I confirm (described by the author)  the difficulty in interpreting ground AOD measurements from the aerosol robotic monitoring networks  (AERONET) and concentrations of PM2.5 near ground level.

The calculation results presented in the i.e. Figure 6, S6, S5, S12 confirm  the usefulness of the proposed approach in exposure assessment of PM2.5 using satellite AOD having massive missing values.

The proposed approach applies to short-term acute incidents, which is very important for human health.

Summary recommendation: publish in present form

Reviewer 2 Report

This manuscript presents an application of using bootstrap aggregating (bagging) of autoencoder-based residual networks to make robust imputation of MAIAC AOD and thereafter PM2.5 estimation at a high spatial (1 km) and temporal (daily) resolution in Jing-Jin-Ji metropolitan region of China.

Some closely related studies have not been referenced. For example, Song et al. (2019) has done a detailed discussion on non-random missingness of AOD over North China Plain and its potential influence on PM2.5 retrieval.

In table 3, its would be more convincing if the performance of original MAIAC AOD (sampling number) against AERONET AOD is added. This can show the improvement of this method in imputation of MAIAC AOD in temporal scale.

As I know, in machine learning, different hyperparameters (mini-batch sizes, activation functions and etc.)  and structures of network would influence the performance of model. So how to ensure certain learning method is fully waken up?

minor errors,

Line 64, 'have been producing' should be 'has'.

Line 285-286, 'and allows' can be 'allow' or ', allows'

Line 82, the reference format of ' Guo et al.' is not consistent with others.

Figure 6b, legend 'Reference (1-1) line' would be zero line.

minor tips,

1. Line 44, ' by coal ' the word 'by' can be omitted.

2. The full names of abbreviations should be given, such as 'NASA' in line 66.

3. 'L1 and L2' in line 289, what are the exact meaning?

Reviewer 3 Report

Comments are given in a separate file.

Round 2

Reviewer 3 Report

I thank the author for his thorough answers to the all points raised in my review, and conducting additional analyses. I disagree with the authors' interpretation of the additional tests (see major point), but find that the manuscript has improved overall and could be published after further amendments, especially an in-depth discussion on the limitations of using the model for spatial extrapolation.

Major point: In Fig. 7 c), additional data are masked, simulating missing data (even though the test is much more limited than what was suggested in my review). The author states that "The performances of the re-trained models
[...] were similar to the models trained using all the samples and the predicted AOD for the additional missing values show reasonable transition between available and imputed MAIAC AOD based on their original observed values." (in lines 425-428). While the test scores may be similar (even though it is not clear to me where to look in Table S2 to find out), substantial differences between predicted MAIAC AOD are apparent in many regions where no data exist to validate the models (cf. northern regions and southwestern region in Fig. 7 b) and d)). As such, the validation technique does not seem appropriate in those situations and can only be used to validate regions where only limited pixels are missing (as in Fig. 6). I think a detailed discussion of the limitations of the technique and the validation is needed in the manuscript.

The last paragraph of the introduction switches in time and should be written in present tense.

line343 This involved matching ...

line346 ... noisy AOD data...

line300 in supplementary: MAIMC --> MAIAC